Psychometric properties of the general self-efficacy scale among Thais with type 2 diabetes: a multicenter study

Hurst Cameron 1 2 3
Rakkapao Nitchamon nitchamonbt@fph.tu.ac.th 2
Malacova Eva 1
Mongkolsomlit Sirima 2
Pongsachareonnont Pear 4
Rangsin Ram 5
Promsiripaiboon Yindee 6
Hartel Gunter 1
1 Biostatistics Group, QIMR Berghofer Medical Research Institute , Brisbane , Queensland , Australia
2 Faculty of Public Health, Thammasat University , Rangsit , Prathumthani , Thailand
3 Molly Wardaguga Research Centre, Charles Darwin University , Brisbane , Queensland , Australia
4 Vitreoretinal Research Unit, Chulalongkorn University , Pathumwan , Bangkok , Thailand
5 Department of Military and Community Medicine, Phramongkutklao College of Medicine , Din Daeng , Bangkok , Thailand
6 Department of Public Health, Surat Thani Rajabhat University , Mueng , Surat Thani , Thailand
Menini Stefano
Electronic publication date: 2022 May 23
Publication date: 2022
Volume: 10
Electronic Location ID: e13398
Received 2021 Oct 29; Accepted 2022 Apr 17
Copyright: ©2022 Hurst et al.
Copyright year: 2022
Copyright holder: Hurst et al.
License: This is an open access article distributed under the terms of the Creative Commons Attribution License, which permits unrestricted use, distribution, reproduction and adaptation in any medium and for any purpose provided that it is properly attributed. For attribution, the original author(s), title, publication source (PeerJ) and either DOI or URL of the article must be cited.
License URL: https://creativecommons.org/licenses/by/4.0/

Keywords: Type 2 Diabetes, Patient self-care, General self-efficacy, Psychometric validation

Funding: Bualuang ASEAN research fellowship for Cameron Hurst 0516.56/2550 This research was supported by a Bualuang ASEAN research fellowship for Cameron Hurst (No. 0516.56/2550). The funders had no role in study design, data collection and analysis, decision to publish, or preparation of the manuscript.

==============================
Background

Type 2 diabetes (T2D) is one of the most common chronic diseases in the world. In recent decades the prevalence of this disease has increased alarmingly in lower to middle income countries, where their resource-limited health care systems have struggled to meet this increased burden. Improving patient self-care by improving diabetes knowledge and diabetes management self-efficacy represents a feasible way of ameliorating the impact of T2D on the patient, and the health care system. Unfortunately, the relationships between self-efficacy, diabetes self-management, and thereafter, patient outcomes, are still far from well understood. Although a domain-specific measure of diabetes management self-efficacy, the Diabetes Management Self-Efficacy Scale (DMSES), has been validated in the Thai T2D population, more general measures of self-efficacy, such as the General Self-Efficacy scale (GSE) have not been validated in this population. In this paper we translate and examine the psychometric properties of the GSE in Thais living with T2D.

Methods

In this nation-wide study we examined the psychometric properties of the GSE in 749 Thais diagnosed with T2D within the last five years, and evaluated its relationship with the DMSES along with other patient characteristics. Reliability of GSE was assessed using Cronbach’s alpha, and the construct validity was examined using confirmatory factor analysis, along with GSE’s convergence and discrimination from DMSES.

Results

The Thai version of the GSE was shown to have good psychometric properties in Thais living with T2D. Cronbach’s alpha was shown to be 0.87 (95% CI [0.86, 0.88]). We also demonstrated the structural validity of the GSE (Tucker-Lewis Index = 0.994, Cumulative Fit Index = 0.995, Adjusted Goodness of Fit Index = 0.998, Root Mean Square Error of Approximations = 0.025, 95% CI [0.06–0.039]), and that this instrument has a similar structure in Thais as in other populations. GSE was also shown to have some overlap with the DMSES with correlations among GSE and the DMSES domains ranging from 0.18 to 0.26, but also the GSE has substantial discrimination from DMSES (Disattenuated correlation coefficient = 0.283, 95% CI [0.214–0.352], p < 0.001). This suggests that while general and diabetes management self-efficacy are somewhat associated, there are aspects of diabetes management self-efficacy not captured by the more stable general self-efficacy.

Conclusions

We demonstrate that the Thai GSE is a reliable and valid measure. We believe the GSE may represent a useful tool to examine the efficacy of proposed and existing diabetes self-management, and management self-efficacy interventions.

Introduction

Diabetes is widely acknowledged as one of the fastest growing chronic conditions worldwide with 1 in 11 adults (463 million) living with the condition, almost 80% of whom live in low or middle income countries (International Diabetes Federation, 2019). A large majority of people living with diabetes in the world (approximately 90%) have Type 2 diabetes(T2D) which is widely considered a lifestyle disease, and factors such as urbanization and increasingly sedentary lifestyles have been strongly linked to the increase of T2D (Hurst, Rakkapao & Hay, 2020). In Thailand, a country with a resource-limited health care system, the prevalence of T2D among adults has tripled over a 25 year period, rising from 2.3% in 1991 to 8.0% in 2015 (Euswas et al., 2021). This also represents a considerable burden on the Thai health care system and economy which is estimated to represent 21% per capita gross domestic product (Chatterjee et al., 2011).

One way to address this increasing burden of T2D is to augment the health care system by facilitating patient self-care. Diabetes is a disease where patients’ conscientious attention to their own disease can prolong, and even prevent, the onset of chronic diabetes complications such as diabetic retinopathy, nephropathy, neuropathy and peripheral artery disease. Indeed it has been suggested that strong diabetes self-management may be almost as efficacious as oral hypoglycemic agents in controlling blood glucose (McDowell et al., 2005), a clinical target widely acknowledged as being highly associated with preventing the onset of downstream chronic diabetes complications. It has been shown that among Thais living with T2D, each unit higher a patient scores on the diabetes self-care scale (when considered as a 10-point scale) doubles the odds of blood glucose control (Hurst, Rakkapao & Hay, 2020).

Self-care in those with diabetes has been shown to be related to diabetes management self-efficacy, the extent to which a patient believes they can manage their condition (McDowell et al., 2005). Diabetes management self-efficacy can be measured by the Diabetes Management Self-Efficacy Scale (DMSES), a well-established scale that has been validated in several populations including Thais living with T2D (Sangruangake, Jirapornkul & Hurst, 2017), as well as those in many other countries including the Dutch (Van der Bijl, Van Poelgeest-Eeltink & Shortridge-Baggett, 1999), Italians (Messina et al., 2018) and Australians (McDowell et al., 2005). The level of a patient’s diabetes management self-efficacy can have a profound effect on patient outcomes. Hurst, Rakkapao & Hay (2020) demonstrated that not only is diabetes management self-efficacy a major mediator of diabetes self-care in terms of controlling blood glucose, but it can even have an additional (direct) effect on blood glucose control. The reason why diabetes management self-efficacy may improve blood glucose control beyond its influence on diabetes self-care is unclear. We need to look more closely at mechanisms underpinning disease management self-efficacy.

The notion of self-efficacy was first proposed by Bandura (1977) and, in general, represents a person’s belief in their ability to perform particular tasks or practices. Importantly, Bandura believed that self-efficacy is a primary predictor of individuals’ behavior change, and people who show the highest levels of behavior change are those with high self-efficacy (Dehghan et al., 2017). Bandura speculated self-efficacy to be strictly task or domain specific (Bandura, 1986; Van der Bijl, Van Poelgeest-Eeltink & Shortridge-Baggett, 1999), whereas others believe that self-efficacy can be generalized to encompass a broader and stable sense of an individual’s personal competence to deal with a range of stressful circumstances (Schwarzer & Jerusalem, 1995; Sherer et al., 1982; Luszczynska, Scholz & Schwarzer, 2005). For instance, if an individual has high self-efficacy in one domain, it might be expected that they are more likely to have high self-efficacy in another. In this respect, self-efficacy can be conceptualized as having both a general (or global) and domain-specific component (Salsman et al., 2019). General self-efficacy may be considered a more stable personality trait and likely to be reflected by someone’s optimism or resilience, whereas more domain-specific aspects of self-efficacy (e.g., diabetes management self-efficacy) may drive more specific aspects of health behavior and therefore more likely to reflect a person’s particular experiences in this more narrowly defined domain (Salsman et al., 2019).

How domain-specific and general self-efficacy instruments converge and diverge may also provide insights into how to best craft intervention components in terms of domain-specific or general aspects of patient empowerment. For example, Luszczynska, Scholz & Schwarzer (2005) showed that GSE had varying levels of association with various domain-specific types of self-efficacy among different populations. For instance, swimming self-efficacy among swimmers was shown to be quite highly associated with the GSE, whereas smoking abstinence self-efficacy had only a weak association with the GSE. This might also flow through to sub-domains of domain-specific self-efficacy instruments. For instance, the extent to which general self-efficacy relates to the various sub-domains of diabetes management self-efficacy (diet-, exercise-, monitoring- and medical treatment self-efficacy) may provide some insights into how both general- and diabetes management self-efficacy are likely to act on patient self-care and patient outcomes, and furthermore, may shed some light on which aspects are more likely to be successfully modified through intervention. For example, in patients with poorer general self-efficacy, it may be easier to influence their adherence to a program of regular blood sugar monitoring than affecting major lifestyle changes such as dietary and exercise behavior.

The GSE is a widely used instrument and has been validated in many populations across the world. While it has been validated in some Asian populations including the Chinese (Zhang, Zhan & Liu, 2018) and Indonesians (Putra, Rahayu & Umar, 2019), no Thai language version of the GSE exists. The aim of this study is to demonstrate the validity of the General Self-Efficacy scale (GSE) in Thai adults with T2D. We then examine its psychometric properties including both its overlap (convergence) and discrimination (divergence) from the domain-specific Diabetes Management Self-Efficacy Scale (DMSES).

Methods

Design and sample

In this multi-center, methodological study, patients from four regions of Thailand (Central, North, North-East and South) were recruited from community hospitals. Patients were sampled using a proportional stratified approach where the appropriate proportions of patients to collect from each region were estimated based using the regional distribution of T2D patients in Thailand’s National Health Security Office’s (NHSO) DMHT (Diabetes and Hypertension) dataset which contains over 300,000 diabetes patients collected from across Thailand over the last nine years. In our study 749 patients, who had been diagnosed with T2D within the last five years, were collected from 12 community hospitals. We obtained ethical approval for the current study from the Thammasat University HREC (ID: 168/2562) and all patients provided written informed consent prior to participation. The informed consent form is provided in the Supplemental Information.

Measurements

Translational validity

We translated the two self-efficacy instruments, the General Self-Efficacy scale(GSE) and the Diabetes-Management Self-Efficacy Scale (DMSES) from English into Thai using the forward and backward translation technique first advocated by Brislin (1970). Specifically, we recruited four Thai-English bilingual translators, the first two of which were asked to translate the original English version of the instruments into Thai (forward-translation), and the remaining two translators were then requested to independently translate these new Thai versions back to English (back-translation). Two native English speakers then compared the original instruments to the back-translated versions. We then field-tested the full survey instrument in a pilot group from the study population (20 Thais living with T2D) to assess the translational quality and the practical aspects of survey administration. In this last part of the process, we asked participants to both read and listen to each item to ensure their understanding. For both the GSE and DMSES we obtained permission from the original developers before undertaking any analysis.

Validation of the DMSES in Thai people with type 2 diabetes

The DMSES was originally developed for use in Western populations (Van der Bijl, Van Poelgeest-Eeltink & Shortridge-Baggett, 1999) and assesses the confidence of diabetes patients in their ability to manage their diet, blood sugar, physical exercise and medical treatment. DMSES has also been shown to be both reliable (Cronbach’s alpha = 0.89) and valid in the Thai T2D population (Sangruangake, Jirapornkul & Hurst, 2017). The DMSES has four domains, diet self-efficacy, exercise self-efficacy, monitoring self-efficacy and regimen self-efficacy, and all items of the DMSES are measured on a five-point ordinal scale ranging from 1 = “Least confident” to 5 = “Most confident”. Sangruangake, Jirapornkul & Hurst (2017) not only showed that DMSES was valid in Thai adults with T2D, but the overall scale and several of its subscales were highly predictive of concurrent HbA1c control. For instance, the overall DMSES was shown to have a sensitivity and specificity for predicting HbA1c control of 81% and 84%, respectively (Sangruangake, Jirapornkul & Hurst, 2017).

The general self-efficacy scale

In this study we will consider the General Self-Efficacy Scale which was first developed in German by Schwarzer & Jerusalem (1995), but then translated, and subsequently validated, for English speaking adults from the United States (Salsman et al., 2019). The GSE has also been shown to be both reliable (Cronbach’s alpha = 0.87) and valid in a T2D population (Qiu, Huang & Wang, 2020). The 10 items in the GSE are on a five-point ordinal scale scored in a similar way to the DMSES above (ranging from 1. “I am not at all confident” to 5. “I am very confident”.). The original developers of GSE, and many since, have demonstrated that the GSE is unidimensional. We obtained permission from the original developers of the GSES.

Other patient characteristics

In addition to the DMSES and GSE, demographic and clinical measurements were collected for each patients. Demographic variables collected included sex, age, BMI, education and income, along with a family history of T2D. Clinical data included HbA1c level, systolic and diastolic blood pressure, LDL-cholesterol, HDL-cholesterol and triglyceride levels. We also recorded chronic diabetes complications and other common comorbidities. Further information about the variables collected as part of this study can be found in the Patient and Medical record Case Report Forms are provided in Hurst (2021).

Statistical analysis

All patient characteristics were summarized using descriptive statistics. For continuous patient characteristics we used means and standard deviations, whereas categorical characteristics were summarized using counts and percentages. In terms of assessing the reliability and validity of the GSE, and within the constraints our study design, we followed the COSMIN guidelines (Mokkink et al., 2010) for establishing psychometric validity. Internal consistency reliability was evaluated using Cronbach’s alpha and, as this is an applied research setting, an alpha over 0.8 was assumed to represent acceptable internal consistency (Nunnally & Bernstein, 1994). Exploratory factor analysis was conducted using Principal Components Analysis followed by a parallel analysis to determine the number of domains for the GSE in the Thai T2D population. Specifically, parallel analysis is used to identify how many principal factor can be extracted, where a principal factor is one that can explain significantly more variation than any single item. The parallel analysis revealed only a single principal factor existed (see Fig. S1), consequently a one factor model was assumed to be sufficient to represent the GSE structure, and any further exploratory factor analysis to elucidate factor structure was deemed unnecessary. Unweighted least squares Confirmatory Factor Analysis was then conducted to assess the structural validity of the resulting measurement model. We used the Tucker-Lewis index (TLI), Cumulative-fit index (CFI), adjusted goodness-of-fit index (AGFI) and root-mean-square error of approximation (RMSEA) to gauge measurement model fit. A model with TLI, CFI, (Hu & Bentler, 1999) GFI (Shelvin & Miles, 1998) and AGFI (Byrne, 2006) ≥ 0.9, and RMSEA ≤ 0.08 (Browne & Cudeck, 1993) was deemed to represent adequate model fit. Despite being widely known to be a poor measure of measurement model fit (Stallman & Hurst, 2016), we also included the χ2 goodness of fit statistic for reasons of convention. Bartlett’s test of sphericity and the Kaiser-Meyer-Olkin (KMO) measure of sampling adequacy were generated along with the CFA to provide further evidence of construct validity (Kaiser, 1974; Kline, 2000). Divergent validity between GSES and DMSES was evaluated using disattenuated correlation analysis where the effect of each scale’s unreliability is removed prior to assessing scale correlation. Disattenuated correlation significantly lower then 0.85 was considered evidence of scale discrimination (Shaffer, DeGeest & Li, 2016). Convergence between GSE and DMSES (and its subscales) was investigated using Pearson’s correlation. Finally, we also wanted to investigate how both GSE and DMSES varied in the Thai T2D population. To examine this we collapsed both GSE and DMSES into three ordered categories (low: x < mean − 1sd, moderate: mean − 1sd ≤ x < mean + 1sd, high: x ≥ mean + 1sd) and investigated the association of patient characteristics with both GSE and DMSES using proportional odds ordinal logistic regression. All analysis was conducted in the R statistical package (v4.0.3) (R Core Team, 2021), with the confirmatory factor analysis being conducted using the R library Lavaan (Rosseel, 2012). A significance level of 0.05 was used throughout all inferential testing.

Results

Patient characteristic are provided in Table 1.

Table 1 Patient characteristics: Sociodemographic and clinical characteristics of the 749 participants.

Patient characteristics	Overall	
Total number of patients	749	
Region (%)		
Central	251 (33.5)	
South	118 (15.8)	
North	144 (19.2)	
North-east	236 (31.5)	
Rural (%)	423 (56.5)	
Age (mean (SD))	53.6 (21.6)	
Female (%)	434 (57.9)	
BMI Class (%)		
Underweight	17 (2.3)	
Normal	301 (41.0)	
Overweight	266 (36.2)	
Obese	151 (20.5)	
Waist Circumference in cm (mean (SD))	89.8 (12.3)	
Years Since T2D Diagnosis (mean (SD))	2.9 (1.9)	
Marital Status (%)		
Single	66 (9.0)	
Married	593 (80.8)	
Widowed, Separated or Divorced	75 (10.2)	
Education (%)		
No formal education	31 (4.1)	
Primary	389 (51.9)	
Secondary	252 (33.6)	
Bachelor’s	67 (8.9)	
Masters and above	10 (1.3)	
Religion (%)		
Buddhist	703 (96.8)	
Muslim	19 (2.6)	
Other	4 (0.6)	
Monthly Income in Thai Baht (%)		
<5,000 (160USD)	173 (23.1)	
5,000–9,999 (160–320 USD)	236 (31.5)	
10,000–14,999 (320–480)	161 (21.5)	
15,000–19,999 (480–640 USD)	79 (10.5)	
20,000–24,999 (640–760 USD)	41 (5.5)	
>25,000 (>760 USD)	59 (7.9)	
Family History of Diabetes (%)	341 (45.5)	
Any Comorbidity (%)	541 (72.3)	
Hypertension (%)	368 (49.1)	
Dyslipidemia (%)	262 (35.0)	
Heart Disease (%)	26 (3.5)	
Chronic Kidney Disease (%)	9 (1.2)	
Other Comorbidity (%)	42 (5.6)	
Diabetes Treatment (%)		
None	25 (3.3)	
Oral hypoglycemic agent (OHA)	660 (88.1)	
Insulin	17 (2.3)	
OHA + Insulin	47 (6.3)	
Smoking (%)		
Never	614 (82.0)	
Previous	80 (10.7)	
Current	55 (7.3)	
Alcohol Use (%)		
Never	564 (75.3)	
Previous	97 (13.0)	
Current	88 (11.7)	
Previous Eye Check (patient reported) (%)	610 (81.8)	
Frequency of Eye Check (patient reported) (%)		
<once per year	29 (4.7)	
Once per year	557 (89.8)	
> once per year	34 (5.5)	
HbA1c <7% (%)	248 (39.6)	
SBP < 140 mmHg & DBP < 80 mmHg B (%)	229 (31.1)	
LDL cholesterol < 100 mg/dL (%)	296 (41.1)	
Notes.

Data are summarized by n (%) for categorical variables, and mean (SD) for continuous variables.

The average age of patients was 53.6 years old (sd = 21.6) with almost 58% of patients being women. Over half of the patients resided in rural areas (56.5%), and a majority of patients were married (80.8%), had primary or secondary school as their highest level of education (85.5%) and earned less than 15,000 Thai Baht (USD$480) per month (76.1%). Almost half of the patients had a family history of diabetes (45.5%) and 72.3% of them had some comorbidity with levels of hypertension and dyslipdemia being particularly high (49.1% and 35% respectively). The levels of blood glucose, blood pressure and low density cholesterol control was 39.6%, 31.1% and 41.1%, respectively.

Reliability and structural validity of the General Self-Efficacy scale

Justification to fit a measurement model (and to provide partial evidence of construct validity) was supported by the Kaiser-Meyer-Olkin statistic demonstrating sampling adequacy (KMO = 0.91) and Bartlett’s test of sphericity indicated that sufficient association exists among the GSE items to warrant fitting a measurement model (χBartlett2=3193.4, df = 45, p < 0.001). Parallel analysis demonstrated that only one factor provided an eigenvalue significantly higher than 1 (see Fig. S1), suggesting a single factor measurement model is the most appropriate fit to the data. Confirmatory factor analysis demonstrated that the GSE does have good structural validity (TLI = 0.994; CFI = 0.995; AGFI = 0.9986; RSMEA = 0.025, 95% CI [0.06–0.039]). Unsurprisingly, the corresponding χ2 test was statistically significant (χ2 = 51.28, df = 35, p < 0.05). The structure and individual standardized item loadings from the GSE measurement model are provided in Table 2. All item loadings were identified as statistically different from zero (all p < 0.001). We also found the GSE to have strong internal consistency reliability with a Cronbach’s alpha of 0.87 (95% CI [0.86, 0.88]).

Table 2 Items of the General Self-Efficacy Scale (GSES) along with their standardized loadings from the confirmatory factor analysis and the Item mean, median and inter-quartile range.

Item	Loading	Mean	Median	
1: I can manage to solve difficult problems if I try hard enough.	0.594	3.242	3	
2: If someone opposes me, I can find the means and ways to get what I want.	0.324	2.867	3	
3: It is easy for me to stick to my aims and accomplish my goals.	0.618	3.172	3	
4: I am confident I can deal efficiently with unexpected events.	0.642	3.105	3	
5: Thanks to my talents and skills, I know how to handle unexpected situations.	0.672	3.092	3	
6: I can solve most problems if I try hard enough.	0.723	3.207	3	
7: I stay calm when facing difficulties because I can handle them.	0.751	3.182	3	
8: I stay calm when facing difficulties because I can handle them.	0.729	3.163	3	
9: If I am in trouble, I can think of a solution.	0.755	3.175	3	
10: I can handle whatever comes my way.	0.741	3.218	3	

Discrimination and convergence between general and diabetes management self-efficacy

Figure 1 shows the correlation of GSE with the total DMSES scale and its subscales and suggests there is substantial overlap among most of the DMSES subscales, although monitoring self-efficacy was only moderately correlated with the other DMSES subscales. In contrast, associations of GSE with the total DMSES scale and the DMSES subscales were considerably lower (correlations ranging from 0.181 to 0.258) providing some evidence of divergence. However, Pearson’s correlation coefficient does not account for measurement error in scales, so we used the disattenuated correlation coefficient (a measure of association between scale that accounts for their unreliability). Disattenuated correlation analysis between GSE and the DMSES provided strong evidence of discriminant validity (rdisattenuation = 0.283, 95% CI [0.214–0.352], p < 0.001).

Figure 1 Corrgram of general- and diabetes management self-efficacy(sub)scales.

The upper triangle provides estimates of Pearson’s correlation coefficient along with their 95% confidence intervals, the lower triangle provides the scatter plots illustrating the pairwise relationships among the various scales, and the main diagonal provides the densities showing the distribution of each scale.

The association of patient characteristics with general and diabetes management self-efficacy

Table 3 shows the unadjusted associations of patient characteristics with both GSE and the overall DMSES.

Table 3 Associations of patient characteristics with general self-efficacy and diabetes management self-efficacy class (low, moderate, high) class from a proportional odds logistic regression.

Effect	General self-efficacy	Diabetes management self-efficacy	
	OR	95% CI	OR	95% CI	
Region (Ref: Central)	χLRT2=90.231, df = 3, p < 0.001	χLRT2=65.901, df = 3, p < 0.001	
South	0.956	(0.583, 1.568)	2.225**	(1.371, 3.61)	
North	2.169**	(1.368, 3.439)	1.512****	(0.965, 2.368)	
North-east	2.341***	(1.563, 3.507)	5.066***	(3.338, 7.688)	
Rural	0.853	(0.62, 1.174)	0.396***	(0.285, 0.549)	
Age	0.999	(0.992, 1.006)	1.000	(0.993, 1.007)	
Female	0.902	(0.656, 1.239)	0.792	(0.58, 1.081)	
Body Mass Index (Ref: Normal)	χLRT2=10.756, df = 3, p = 0.013	χLRT2=2.585, df = 3, p = 0.460	
Underweight	0.388****	(0.14, 1.073)	1.335	(0.488, 3.652)	
Overweight	1.151	(0.8, 1.655)	1.065	(0.747, 1.518)	
Obese	0.613*	(0.395, 0.952)	0.779	(0.515, 1.179)	
Waist Circumference	0.988****	(0.975, 1.002)	0.997	(0.984, 1.011)	
Duration of T2D	1.013	(0.933, 1.101)	1.066	(0.984, 1.155)	
Marital Status (Ref: Single)	χLRT2=2.287, df = 2, p = 0.239	χLRT2=0.175, df = 2, p = 0.916	
Married	1.450	(0.816, 2.576)	0.926	(0.535, 1.604)	
Widowed/Separated/Divorced	1.878	(0.904, 3.903)	1.011	(0.5, 2.045)	
Education (Ref: No formal education)	χLRT2=5.919, df = 4, p = 0.205	χLRT2=26.808, df = 4, p < 0.001	
Primary	1.778	(0.808, 3.913)	2.566***	(1.225, 5.374)	
Secondary	1.754*	(0.784, 3.928)	2.386**	(1.12, 5.081)	
Bachelor’s	2.806	(1.114, 7.067)	7.462***	(3.113, 17.888)	
Masters or above	3.479	(0.754, 16.052)	8.964**	(2.169, 37.05)	
Religion (Ref: Buddhist)	χLRT2=0.703, df = 2, p = 0.704	χLRT2=0.064, df =, p = 0.968	
Muslim	1.222	(0.474, 3.149)	1.118	(0.443, 2.822)	
Other	2.139	(0.296, 15.447)	1.118	(0.119, 10.503)	
Monthly Income (Ref: <5000THB)	χLRT2=20.799, df = 5, p < 0.001	χLRT2=48.943, df = 5, p < 0.001	
5,000–9,999THB	1.269	(0.82, 1.965)	0.796	(0.519, 1.222)	
10,000-14,999 THB	0.994	(0.617, 1.603)	1.174	(0.734, 1.878)	
15,000–19,999 THB	1.796****	(0.988, 3.264)	2.113*	(1.182, 3.777)	
20,000–24,999 THB	2.467*	(1.181, 5.15)	2.078****	(0.995, 4.342)	
>25,000 THB	3.299***	(1.757, 6.195)	5.697***	(3.078, 10.544)	
Family History of Diabetes	1.212	(0.883, 1.663)	1.758***	(1.285, 2.405)	
Any Comorbidity	0.761	(0.536, 1.08)	0.967	(0.688, 1.36)	
Hypertension	0.959	(0.7, 1.314)	0.928	(0.683, 1.26)	
Dyslipidemia	0.734****	(0.526, 1.023)	0.808	(0.586, 1.114)	
Heart Disease	0.692	(0.298, 1.607)	1.107	(0.48, 2.556)	
Chronic Kidney Disease	0.531	(0.136, 2.069)	1.819	(0.412, 8.032)	
Other Comorbidity	0.808	(0.406, 1.609)	0.741	(0.379, 1.45)	
T2D Treatment (Ref: Diet and Exercise)	χLRT2=1.148, df = 3, p = 0.766	χLRT2=2.434, df = 3, p = 0.487	
Oral hypoglycemic agents (OHTs)	0.889	(0.368, 2.145)	0.771	(0.324, 1.833)	
Insulin	0.711	(0.184, 2.745)	1.312	(0.359, 4.798)	
OHT + Insulin	0.654	(0.227, 1.886)	0.570	(0.201, 1.615)	
Smoking (Ref: Never)	χLRT2=2.279, df = 2, p = 0.248	χLRT2=13.188, df = 2, p = 0.001	
Previous	1.456	(0.88, 2.408)	2.336***	(1.429, 3.818)	
Current	1.336	(0.745, 2.398)	1.675****	(0.928, 3.024)	
Alcohol Use (Ref: Never)	χLRT2=3.967, df = 2, p = 0.138	χLRT2=14.459, df = 2, p < 0.001	
Previous	1.200	(0.748, 1.926)	1.945**	(1.23, 3.074)	
Current	1.612****	(0.996, 2.609)	2.057**	(1.268, 3.338)	
Notes.

* p < 0.05.

** p < 0.01.

*** p < 0.001.

**** p < 0.1.

THB Thai Baht

Provincial locations tended to be associated with higher self-efficacy relative to those living in the more highly developed central region, although the patterns differed a little between the general and domain-specific self-efficacy measures. For both types of self-efficacy, those living in the north-east had considerably higher odds of better self efficacy (ORGSE = 2.341; 95% CI [1.563, 3.507]; p < 0.001 and ORDMSES = 5.066; 95% CI [3.338–7.688]; p < 0.001) whereas those in the North were only better with GSE (ORGSE = 2.169; 95% CI [1.368–3.439]; p < 0.01) and those in the South, only better with DMSES (ORDMSES = 2.225; 95% CI [1.371–3.613]; p < 0.01). Income also had similar patterns across both scales with those in the higher income brackets having higher odds of better general and diabetes management self-efficacy (Table 3). This pattern in effect sizes was also observed for education where there seemed to be a quite strong dose–response effect; the higher the level of eduction, the better the odds of higher self-efficacy. However, despite these large effects, statistical significance could only be established for DMSES (Table 3). Somewhat counter-intuitively, current alcohol use was associated with higher self-efficacy of both types, albeit only a trend for GSE (ORGSE = 1.612; 95% CI [0.996, 2.609]; p < 0.1; ORDMSES = 2.057; 95% CI [1.268–3.338]; p < 0.01). Both previous alcohol use and smoking were associated with diabetes management self-efficacy (ORalcohol = 1.945; 95% CI [1.23–3.074]; p < 0.01; ORsmoking = 2.336; 95% CI [1.429–3.818]; p < 0.001).

Patients that had a family history of diabetes had considerably higher odds of diabetes management self-efficacy (ORDMSES = 1.758; 95% CI [1.285–2.40]; p < 0.001), but there was no such association with general self-efficacy. Similarly, rurality could only be shown to be related with diabetes management self-efficacy, however, in this case rural residence is associated with poorer diabetes management self-efficacy (ORDMSES = 0.396; 95% CI [0.285–0.549]; p < 0.01).

Discussion

In this study we demonstrate the reliability and validity of the GSE in Thais living with diabetes, and explore its psychometric properties. We show that the unidimensional GSE is structurally valid in the Thai T2D population and that it has substantial overlap with the domain-specific DMSES, but it also has sufficient discrimination to measure aspects of global self-efficacy not captured by DMSES. This suggests that there is a considerable proportion of domain-specific diabetes management self-efficacy not explained by a person’s general self-efficacy, and consequently, diabetes management self-efficacy which has been shown to be a strong predictor of blood-glucose control among Thai T2D patients (Sangruangake, Jirapornkul & Hurst, 2017), may be improved by well-designed disease-focused behavioral interventions. Indeed, several studies have trialled interventions aimed at improving diabetes self-care through improving patient diabetes management self-efficacy in other populations, and have shown considerable success (Wong et al., 2015; Lin et al., 2020).

In terms of the correlation of GSE with the DMSES and its subscales, despite being statistically significant, we found only weak to moderate associations between the general and diabetes self-management specific measures of self-efficacy. In our sample we also observed that some domains of DMSES (e.g., Monitoring self-efficacy) correlated more highly with GSE, than others (e.g., Regimen self-efficacy). What this means in terms of the efficacy of current self-efficacy interventions in people living with diabetes, and what insights it might provide in the design of future interventions is something that needs further consideration. However, it does highlight that the relationship between general and different types of domain specific types of self-efficacy may be both subtle and complicated. This is something we see in other studies of the relationship between general self-efficacy with other domain-specific measures of self-efficacy. The magnitude of these associations is comparable to those found in Luszczynska, Scholz & Schwarzer (2005) who conducted a correlation analysis of GSE with other domain-specific measures of self-efficacy such as exercise-, nutrition- and smoking abstinence- self-efficacy. It is also worth noting the internal consistency reliability we found among Thais living with diabetes(0.87) was consistent with that found in other populations such as those from Germany (Schwarzer & Jerusalem, 1995).

In terms of how both GSE and DMSES vary with patient characteristics, we found region of residence, education and income to be the main drivers of both general and diabetes management self-efficacy. Interestingly, Northeastern Thais, a region with the highest prevalence of T2D (Sieng et al., 2015), had a considerably greater chance of higher general and diabetes-management self-efficacy. We also found that alcohol and tobacco use were also associated with both higher GSE and higher DMSES, although there may be a reverse causal association here. Indeed, it brings up the question of whether measures of self-efficacy may also reflect patient complacency. Some of those who feel confident in managing their disease may underestimate the difficulties and/or commitment in doing so.

We demonstrate that the GSE has excellent psychometric properties, and it is suitable for use in Thais living with T2D, and that this measure can now be used not only as either a health outcome or determinant, but we can now explore what aspects of self-efficacy may be amenable to change through intervention, and which might be more closely associated with a patient’s underlying personality traits. In short, we gain a clearer view of what aspects of self-efficacy may be amenable to change through intervention, and which are more likely to be immutable. It is also likely that GSE is likely to be valid in other Thai chronic disease populations especially in diseases that can have a lifestyle etiology such as hypertension and hypercholesterolemia. It is also likely that GSE would be likely to perform well in culturally similar T2D populations such as those of Vietnam, Laos, Cambodia, Malaysia and Myanmar. However, GSE measurement validity would still need to be formally established in these populations.

Although we did take care to obtain a geographically representative sample of the recently diagnosed Thai T2D population in this multi-center study, we only sampled patients from community hospitals which predominately service patients in the earlier stages of the disease, and even where chronic complications are present, morbidity is generally lower in patients attending these hospitals compared to those attending larger tertiary hospital outpatient clinics in Thailand (Sieng et al., 2015). It is also important to note, however, that community hospitals in Thailand service a large majority of T2D outpatients, especially those in the earlier stage of the disease (Sieng et al., 2015), where lifestyle interventions are more likely to be effective. Although our study was initially designed to include larger hospital outpatient clinics, the COVID-19 epidemic occurring at the time of sampling led to institution-specific administrative hurdles making access to their patients very difficult. Finally, given patients were surveyed only once, at their yearly check-up, we were unable to follow the COSMIN guidelines in their entirety. In particular, the cross-sectional nature of our study design meant we were unable to assess test-retest reliability of GSE, nor GSE’s responsiveness or predictive validity.

However, our study also had some major strengths. Our study design resulted in a nationally representative sample of people living with the earlier stages of diabetes in Thailand. As a result we feel our findings strongly confirm that the GSE is generalizable to all Thai’s living with the earlier stages of T2D. In this study we demonstrate that the Thai version of the GSE is psychometrically valid in the Thai T2D population. This patient-centered construct is likely to be useful in teasing out the complex interplay between patient disease knowledge, self-care attitude and practice. This understanding will provide strong guidance in designing well-grounded cognitive behavior interventions which in turn represent a cost-effective way of improving patients disease self-management, and subsequently, better patient outcomes in a resource-limited health care setting.

Conclusions

We demonstrated that the Thai version of the GSE is both a valid and reliable measure of general self-efficacy among Thais living with Type 2 Diabetes. We believe that it is also likely to prove valid among Thais with other metabolic syndrome related conditions such as hypertension and dyslipidemia. Furthermore, we believe that the GSE may be useful in being able to tease out the subtle interplay between disease knowledge, disease management self-efficacy, patient self-care, and how these relate to downstream patient outcomes. Stronger understanding of these relationships is likely to lead to the development of better focused and effective interventions that can help delay, and even prevent, the development of chronic diabetes complications.

Supplemental Information

Figure S1 Scree plot resulting from a Principal Components Analysis-based parallel analysis of GSE

The parallel analysis involves a permutation test to test the hypothesis that each component’s eigenvalue is not greater than 1.

Click here for additional data file.

Additional Information and Declarations

Competing Interests

Author Contributions

Human Ethics

Data Availability

The authors declare there are no competing interests.

Cameron Hurst conceived and designed the experiments, performed the experiments, analyzed the data, prepared figures and/or tables, authored or reviewed drafts of the paper, and approved the final draft.

Nitchamon Rakkapao conceived and designed the experiments, performed the experiments, authored or reviewed drafts of the paper, and approved the final draft.

Eva Malacova, Sirima Mongkolsomlit, Pear Pongsachareonnont, Ram Rangsin and Yindee Promsiripaiboon conceived and designed the experiments, authored or reviewed drafts of the paper, and approved the final draft.

Gunter Hartel conceived and designed the experiments, analyzed the data, authored or reviewed drafts of the paper, and approved the final draft.

The following information was supplied relating to ethical approvals (i.e., approving body and any reference numbers):

Thammasat University Human Research Ethics Committee.

The following information was supplied regarding data availability:

The raw data is available at Mendeley: Hurst, Cameron (2021), “Impact of disease self-care, management self-efficacy and knowledge in Thais with Type 2 diabetes: A 5 year nationwide cohort study ”, Mendeley Data, V1, doi: 10.17632/6hv4g67vb6.1.

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
