# Peer review of "Psychometric properties of the general self-efficacy scale among Thais with type 2 diabetes: a multicenter study"

_PeerJ, doi:10.7717/peerj.13398_

## Round 0.1 · original submission · Major Revisions

Dear Dr. Rakkapao,

Your manuscript entitled “Psychometric properties of the General Self-Efficacy Scale among Thais with Type 2 diabetes, and its relationship with patients' disease management self-efficacy: A multicenter study" which you submitted to PeerJ, has been reviewed by the editor and 3 external reviewers.

I regret to inform you that the reviewers have raised some significant concerns that need to be addressed before the manuscript can be considered further. However, since reviewers do find some merit in the paper, I would be willing to reconsider if you wish to undertake major revisions and resubmit.

If you decide to resubmit the revised version, please summarize all the improvements made in the new version and give answers to all critical points raised in the reviewers’ report in an accompanying letter. Please copy and paste each and every reviewer's comment above your response.

Please carefully consider the comments regarding the study design, methods, and discussion of relevant literature in the field.

Please also note that resubmitting your manuscript does not guarantee eventual acceptance. Since the requested changes are major, the revised manuscript will undergo a second round of review by the same reviewers. I must emphasize that the acceptability of the revision will depend upon the resolution of the points raised by the reviewers.

Sincerely yours,
Stefano Menini

Reviewer 1 ·

Excellent Review

This review has been rated excellent by staff (in the top 15% of reviews)
EDITOR COMMENT
Excellent review comments like this help to enhance the quality of the manuscript. I thank the reviewer for the constructive criticism, concrete suggestions, and experienced advice.

Basic reporting

1.1 Clear, unambiguous, professional English language used throughout.
There are some flaws in the writing. To be specific:
The tenses in the abstract are used inconsistently. The authors use alternatingly present and past tense for the same time level.

There are some typing errors, e.g.
Line 274:
It must be ‘this point’ instead of ‘thise point’.

Some sentences have a strange grammar, e.g.
Lines 61-63:
‘It has been shown that among Thais living with T2D that each additional unit a patient scores on the diabetes self-care scale (when considered as a 10-point scale) doubles the odds of blood glucose control (Hurst et al., 2020).’.
Sometimes the references to the literature are given in a confusing manner e.g. . Lines 68-70:
‘Hurst and colleagues Hurst et al. (2020) demonstrated that not only is diabetes management self-efficacy a major mediator of diabetes self-care in terms of controlling blood glucose, but it can even have an additional (direct) effect on blood glucose control.’
The first bracket must be placed before ‘Hurst et al.’.

1.2 Intro & background to show context. Literature well referenced & relevant.
The claim
‘Self-care in those with diabetes has been shown to have two main antecedents, Diabetes Knowledge, the extent to which a patient is informed about the risks, controls and outcomes of their disease, and Diabetes management self-efficacy, the extent to which a patient believes they can manage their condition.’
must be supported by references to literature.
Except for a reference to Bandura the authors do not discuss any psychological models or theories regarding the prediction and explanation of health-related behaviour. Altogether, the authors’ present introduction does not convey the impression that the authors have a deeper understanding of that part of psychology that is relevant for their paper. I recommend the authors to look into ‘Conner M, Norman P: Predicting and Changing Health Behaviour: Research and Practice with Social Cognition Models. Open University Press: Birkshire, England’. This is an excellent and in depth discussion of such models or theories.

1.3 Structure conforms to PeerJ standards, discipline norm, or improved for clarity.
With one exception, the structure conforms to the standards. The exception is the abstract. This should be explicitly structured into background, methodology, results, and conclusions.

1.4 Figures are relevant, high quality, well. Labelled & described.
The information in both figures should be given in tables. The table corresponding to figure 1 could and should also contain descriptive information regarding the items (mean, median etc…). The table corresponding to figure 2 should contain exact numerical information. The left lower triangle of the matrix is not that important. It could be left out.
In any case, the authors should present a table with the items of the GSES and perhaps a table with those of the DMSES.

1.5 Raw data supplied (see PeerJ policy).
In the Mendeley data repository, the search term ‘Hurst, 2021’ yields 211 entries. I leave it to the editorial team to find out whether one of these entries matches.

Experimental design

2.1 Original primary research within Scope of the journal.
As far as I can judge it, the paper is within the scope of the journal. However, the editorial team can judge this better.
2.2 Research question well defined, relevant & meaningful. It is stated how the research fills an identified knowledge gap.
The research question is not well defined. Different parts of the text give different ideas as to what the research question(s) might be. According to the title, the research questions are to investigate the psychometric properties of the General Self-Efficacy Scale and its relation with patients' disease management self-efficacy. Further parts of the text make clear that the authors operationalise patients' disease management self-efficacy solely with the DMSES and, thereby, factually identify the construct of patients' disease management self-efficacy with the instrument DMSES. In the abstract, the authors state that they examine the reliability and validity of the General Self-Efficacy Scale. When I read the abstract I understood this as a specification of the expression ‘psychometric properties of the General Self-Efficacy Scale’, which the authors use in the title.
The further parts of the paper suggest that the authors do not take the investigation of reliability very serious. In the introduction, they do not use the term ‘reliability’ at all. They only talk about validity. In the methods section they use the term ‘reliability’ again in the sub-chapter ‘Statistical analyses’. They announce that they want to investigate reliability using Cronbach’s alpha. The results section contains a sub-chapter with the title ‘Reliability and structural validity of the General Self-Efficacy scale’. However, the results for Cronbach’s alpha are reported neither in this sub-chapter nor in any other part of the paper. I also do not find any other result in the results section that I can understand as an indicator of reliability. Nevertheless, at the beginning of the discussion, the authors write:
‘In this study we demonstrate the reliability and validity of the GSES in Thais living with diabetes, and explore its psychometric properties.’
This sentence confuses me in several regards. First of all, I am not sure what the authors mean by ‘this study’. If they meant the study reported in the paper to be reviewed, the verb would have to be in past tense. As I cannot image which other study they could mean, I guess that the tense is a linguistic flaw. If it is a linguistic flaw, I am confused about the claim that they have demonstrated reliability. I have not seen that they have demonstrated reliability in the paper. I am also confused because the sentence suggests that the authors seem to understand reliability and validity as something different from psychometric properties. Otherwise, there would be no reason for the second part of the sentence. Counting reliability and validity as something different from psychometric properties contradicts massively to what I have learned in my lessons of test theory. According to these and many different lessons, reliability and validity are the most important psychometric properties. Moreover, it becomes unclear, what the authors mean by psychometric properties if they do not count reliability and validity as psychometric properties.
The concept of validity is treated more extensively in the paper. The authors consider two aspects of validity: 1) an aspect of validity that they refer to interchangeably as construct validity or structural validity and 2) divergent validity. With regard to the first aspect, they investigate whether GSES is one dimensional, with regard to the second whether GSES and DMSES diverge. The authors also refer to a theoretical approach that implies such relationships for self-efficacy (Salsman). With respect to this approach, the investigations proposed by the authors can actually be considered as investigations addressing validity. However, I do not see that these structural features are the most important aspects of general self-efficacy. Many conceptions of general intelligence have the same structural feature, but at least in my opinion, general self-efficacy is not the same as general intelligence. More relevant for judging the validity of GSES are relationships between this measurement instrument and empirical phenomena that can be considered as valid indicators of self-efficacy.
A further research question that, again and again, pops up in the paper is to identify those psychological constructs that should be addressed by interventions in order to finally affect clinical parameters. The authors equate the answer to this question with finding that relationship between GSES and DMSES that I have described in the preceding paragraph. However, equating these two answers makes no sense. The relationship between GSES and DMSES determine neither which of the both variables addressed by theses measurement instruments is easier to affect by an intervention nor which of both variables has the stronger impact on self-management behaviour and on clinical parameters.
Still a further research question regards the relationship between different demographic variables on the one hand and GSES and DMSES. This research question is never explicitly formulated as a research question but pops up, all of a sudden, in the sub-chapter ‘Statistical analyses’ and is treated extensively in the results section. Neither the relationship between the treatment of this research question and the remaining research questions nor the relevance of this research question are made clear.

2.3 Rigorous investigation performed to a high technical & ethical standard.
The different research questions are treated with quality:
The reliability is actually not at all investigated. At least, the results are not reported. However, if the authors had done what they have announced to do in the methods part, i.e. if they had computed and reported Cronbach’s alpha this would have been standard. However, a test-retest study would be a better approach for assessing reliability.
The authors have investigated those aspects of validity that they have addressed with very sophisticated methods. However, these aspects of validity are not the most important aspects. More important are relationships between the measurement instrument and obvious indicators of the construct to be measured.
The research question as to which variables should best be addressed by interventions is in no way answered.
The relationship between different socio-demographic features as predictors and the two self-efficacy measures as criteria is investigated with a method that seems peculiar to me. Each of the two criterion measures is converted into an ordinal three-category scale, and the relationship is analysed using proportional odds ordinal logistic regression. The author do not report why they apply this approach. In my opinion, this approach does not make much sense. Converting the original scales into three-category scales destroys much information. In my opinion, the most appropriate method would be linear regression. This approach would not only make use of more information and, thereby, provide more valid results, but coefficients of the linear regression can also be better interpreted than coefficients of proportional odds ordinal logistic regression. If the reason for the approach chosen by the authors should be concerns regarding heteroscedasticity, I would take weighted least squares.

2.4 Methods described with sufficient detail & information to replicate.
By and large, the methods are sufficiently described. However, some explanations regarding proportional odds ordinal logistic regression would have been useful.

Validity of the findings

3.1 Impact and novelty not assessed. Meaningful replication encouraged where rationale & benefit to literature is clearly stated.
I do not understand the meaning of this point.

3.2 All underlying data have been provided; they are robust, statistically sound, & controlled.
I cannot judge this. Even if I had access to the data, I could not do this. Moreover, I wonder what the editors mean by statistically sound data.

3.3 Conclusions are well stated, linked to original research question & limited to supporting results.
Due to the confusion caused in the preceding parts of the paper, the conclusions are also confusing.

Additional comments

The paper might become acceptable with following modifications:
1) A psychologically more professional treatment of the construct of self-efficacy in the introduction
2) Addition of the results regarding reliability
3) Integration of the investigation of the relationship between the instruments and socio-demopgraphic variables in the investigation of the validity or, if the authors do not know how to this, omission of the investigation of the relationships with the socio-demographic variables.
4) Changing the title into 'Reliability and validty of the General Self-Efficacy Scale among Thais with Type 2 diabetes: A multicenter study’

·

Basic reporting

1. The manuscript “flows”. It is easy to read smoothly from beginning to end with clear connections between ideas. It’s clear and easy to follow. Readers can move easily from one major idea to the next without confusing jumps in the Authors’ train of thought.

2. The Authors conducted a thorough literature review referencing the relevant authors of original concepts, prior studies, including a variety of sources from the last years.

3. Tables and figures are appropriately designed. All relevant statistics are included.
Table 1: repeat row header across other pages

4. All results relevant to the hypothesis.

Experimental design

1. The research is novel and a good addition to the evidence base, addressing a gap in the literature.

2. The submission clearly define the research question.
Investigating the psychometric properties of the translated psychological tools is relevant and meaningful for avoiding the development of artefacts based on studies using inadequately prepared tools.
The Authors identified the knowledge gap and investigated the relationships between more general self-efficacy and diabetes-specific self-efficacy. The idea to separate these constructs is interesting and contribute to the knowledge about important issues for patient self-management skills.

3. No ethical concerns.
The Researchers got permission to use the original tool from Professor Schwarzer and received written informed consents from patients prior to participation.

4. More information about basic psychometric properties of the original version of the General Self-Efficacy Scale and the Thai version of DMSES should be provided (e.g reliability - Cronbach’s alpha)

Validity of the findings

1. The rationale of this study is clearly described.

2. The statistical analysis are adequate, clearly explained and presented.
I have one comment. The Authors stated in the Statistical analysis section that: "Internal consistency reliability was evaluated using Cronbach’s alpha...". However, information about the value of Cronbach's alpha was not provided in the Results section.

3. Conclusions are clearly stated. The authors referred to the aims and hypothesis. Discussion is relevant and coherent with the results.

Additional comments

In general, the paper is very well written. Basically, the only thing missing is the value of Cronbach's alpha.

·

Basic reporting

Clear and unambiguous, professional English used throughout.
The article in general has been written in clear, unambiguous, technically correct English and is conform to professional standards of courtesy and expression. A few remarks:
- Line 54: please write the abbreviation ‘GDP’ in full the first time
- Line 151: ‘gudielines’ must be ‘guidelines’
- Line 178: ‘characteristic’ must be ‘characteristics’

Literature references, sufficient field background/context provided.
In my view the article is lacking sufficient introduction and background to demonstrate how the work fits into the broader field of knowledge. First no reference has been given in line 64 related to ‘have been shown’ that diabetes self-care has two antecedents. Second no elaboration has been given about the arguable concept of general self-efficacy (line 85). Bandura (2006) himself explicitly claims in Chapter 14 Guide for Constructing Self-Efficacy Scales in the book Self-Efficacy Beliefs of Adolescents from F. Pajares, & T. Urdan (Eds.), that ‘the efficacy belief system is not a global trait but a differentiated set of beliefs linked to distinct realms of functioning’ (p. 307). However, the authors state that ‘General self-efficacy may be considered a more stable personality trait’ (line 78) referring to Salsman et al., but without any foundation. Third line 84 is unclear: ‘…..importantly what aspects are more likely to be successfully modified through intervention.’ Fourth the aim of the study is ‘to demonstrate the validity (and reliability? - reviewer) of the General Self-Efficacy Scale (GSES) in Thai adults with T2D’ (lines 92 & 93). Why then also examining the relationship between general self-efficacy and diabetes management self-efficacy as this is already part of demonstrating the construct validity of the GSES?
Literature is not appropriately referenced: line 121 (Bijl et al., 1999) must be (van der Bijl et al., 1999) and the matching reference in the reference list (line 304) needs to be changed into: Bijl, J. van der, Poelgeest-Eeltink, A. van, and Shortridge-Baggett, L. (1999). And in line 122 the domain of medical treatment is missing.

Professional article structure, figures, tables. Raw data shared.
Conclusions are missing. Figures, tables and raw data of the article look okay.

Self-contained with relevant results to hypotheses.
The submission should not treat the relationship between GSES and DMSES as a separate aim of the study, because - as stated earlier - it is part of the construct validity testing process of the GSES. Other psycho/clinimetric instruments were included but will be used in future work. This is not relevant information. Don’t mention it.

Experimental design

Research question well defined, relevant & meaningful. It is stated how research fills an identified knowledge gap.
The primary aim of the study must be the question about the validity and reliability of the Thai GSES, because there is no translated and validated Thai version of the GSES available. The argumentation for the value of and using the GSES in the Thai diabetes population must come from the study The General Self-Efficacy Scale: Multicultural Validation Studies by Luszczynska, Scholz, and Schwarzer in The Journal of Psychology, 2005, 139(5), 439–457.

Rigorous investigation performed to a high technical & ethical standard.
The research has been conducted in conformity with the existing technical- and ethical standards in the field.

Methods described with sufficient detail & information to replicate.
Methods section need more focus on the psychometric properties of the GSES. First the design is not a prospective, longitudinal cohort study. Although, data are collected during several years, the analysis is cross-sectional. At one point in time the collected data were analysed. A more appropriate description of the design is a methodological design looking at the psychometric properties of the GSES or a psychometric study. Second the COSMIN-validation methods were not completely followed. For example, the test-retest reliability of the GSES is missing and also the longitudinal validity (responsiveness) of the GSES has not been checked. This was one of the questions stated in the introduction: ‘how modifiable may patients’ attitude towards their disease self-management be?’ Thirdly it is not clear why the translation validity of the DMSES has been checked again, while Sangruangake et al. already showed in 2017 that the DMSES was valid in Thai adults with T2D.

Validity of the findings

Impact and novelty not assessed. Meaningful replication encouraged where rationale & benefit to literature is clearly stated.
Rationale and benefit to literature is not clearly stated. Explicitly include the results of the study of Luszczynska et al. (2005).

All underlying data have been provided; they are robust, statistically sound, & controlled.
Data have been provided, and the statistical analyses are sound and relevant.

Conclusions are well stated, linked to original research question & limited to supporting results.
There are no explicit stated conclusions. The Discussion elaborates the results and limitations but clear conclusions are not given.

Additional comments

No comment

---

## Round 0.2 · Major Revisions

Dear Dr. Rakkapao,

Your manuscript entitled "Psychometric properties of the General Self-Efficacy Scale among Thais with Type 2 diabetes, and its relationship with patients' disease management self-efficacy: A multicenter study " has again been carefully reviewed by the Editor and Reviewers.

Two Reviewers still raise several issues that need to be addressed.

If you decide to resubmit a re-revised version of your manuscript, please summarize all the improvements made in the new version and give answers to all critical points raised in the reviewers’ report in an accompanying letter. Please copy and paste each and every reviewer's comment above your response. If you feel any of their points are inappropriate, you are certainly free to provide rebuttal in your covering letter.

Please note that resubmitting your manuscript does not guarantee eventual acceptance. I reiterate that the acceptability of the revision will depend upon the resolution of all the points raised by the reviewers.

Sincerely yours,
Stefano Menini

Reviewer 1 ·

Basic reporting

GENERAL REMARK: The information regarding line numbers refers to the version with changes in tracking mode.

1 Basic reporting
1.1 Clear, unambiguous, professional English language used throughout
There are still several flaws in basic reporting. The most essential flaws refer to the introduction. The authors have incorporated their reactions to the review comments in such a manner that the argumentation flow is destroyed. This refers to lines 67 to 71 (‘The diabetes…’ to ‘…(McDowell et al., 2005).’) and to lines 99 to 115 (‘Luszczynska…’ to ‘…GSES exists.’) Moreover, the latter text part is not well structured in itself and references are missing.

In addition to this, there are still further minor flaws

Lines 26 to 28
The authors write
‘Improving patient self-care through its antecedents, diabetes knowledge and diabetes management self-efficacy, represents a feasible way of ameliorating the impact of T2D for both the patient, and in health care systems.‘
I cannot make sense of this sentence. There seems to be a linguistic flaw.

Line 84
There must be an apostrophe, i.e. it must be ‘an individual’s’.

Line 171
There is a problem with the reference. It should be placed in front of the full stop. Moreover, in this case, reporting the publication year would do.
Lines 179-180: This sentence is poorly formulated. Very confusing.
Lines 194-195: ‘being well established as a poor measure’ is a very peculiar formulation.
Lines 278 to 280:
The authors write:
‘We also observed that in our sample that some domains of DMSES (e.g. Monitoring Self-efficacy) correlated more highly with GSES, then others (e.g. Regimen DMSEs).’
I suggest writing:
‘In our sample we also observed that some domains of DMSES (e.g. Monitoring Self-efficacy) correlated more highly with GSES, than others (e.g. Regimen DMSEs).’

Line 291
The authors write:
‘In terms of how both GSES and DMSES vary with patient characteristics. We found region of…’
There must be a comma behind ‘characteristics’.

Lines 296-298
The authors write:
‘Indeed, it brings up the question of whether measures of self-efficacy may also reflect patient complacency; Those who feel confident in managing their disease may include individuals who underestimate the difficulties and/or commitment in doing so.’
There must be a full stop behind ‘complacency’. Moreover, the second sentence is difficult to understand. I suggest writing:
‘Some of those who feel confident in managing their disease may underestimate the difficulties and/or commitment in doing so.’

1.2 Further flaws in presentation or, respectively, minor flaws in argumentation
Lines 85-86
What does it mean that something seems logical? From an epistemic point of view, substantiating an empirical claim by stating that this seems logical is a bit naive.
Line 87
I guess there is more than one specific domain. In this case, there should also be more than one domain-specific component of self-efficacy.

Line 121
The authors follow the suggestion of reviewer 3 to call their study ‘methodological’. I think that this advice of reviewer 3 was not wise. There is no study design called ‘methodological study’. The authors should use the other term mentioned by reviewer 3, i.e. cross-sectional study.
1) The authors should divide the whole sub-chapter into more than one paragraph.
2) Before describing a statistical procedure, the authors should mention the purpose for which this procedure is applied. Especially, this includes giving a clear overview of the criteria applied for judging validity. Still more especially, this includes some information they have provided in their comments to my previous review.
3) A part of the text within this sub-chapter already refers to results. The authors should formulate their methods part so that there is no need to refer to results and present the results in the results part.

Experimental design

2 Experimental design
The research question are now much formulated than in the first version. However, there is one commentary of mine that the authors have not adequately addressed. I assume that the reason is that my commentary in the previous review was not clear enough. Therefore, I want to reformulate it now:
‘The interrelation between two variables gives no information as to which of these two variables can be better affected by an intervention!’
The authors should omit all considerations into this direction!

In addition to this, I have remarks regarding Sub-sub-chapters ‘Validation of the DMSES in Thai people with Type 2 Diabetes’ and ‘The General Self-Efficacy Scale’ and the sub-chapter ‘Statistical Analyses’.
Sub-sub-chapters ‘Validation of the DMSES in Thai people with Type 2 Diabetes’ and ‘The General Self-Efficacy Scale’
1) In both cases, the authors should report the criteria that have been applied for testing validity (lines 151 and 161).
2) In both cases, the authors assign ordinal scale level to the answer categories of the basic items. However, all statistical procedures with which the authors process the data of these items require interval scale level. This even applies to Cronbach’s alpha. For ordinal scales, the results of all these statistical procedures would be empirically meaningless. I urgently recommend omitting the word ‘ordinal’ in lines 154 and 163. Otherwise, the authors would have to redo all their analyses with statistical procedures that are appropriate for ordinal scales.
3) The authors should report in both cases how the items are aggregated to values for the total instrument. By the way, for items with interval scale level, summing up would be ok. For items with ordinal level, some kind of IRT-scaling would be necessary.

Sub-chapter ‘Statistical Analyses’
I can understand the authors reply to my comment regarding categorizing the self-efficacy-scales and then computing proportional odds ordinal logistic regression. In reaction to the authors’ reply, I would prefer standardized beta. However, I do not want to be overly dogmatic. So, if the authors feel better with this they should stick to their procedure.

Validity of the findings

3 Validity of findings
All conclusions regarding the question as to which of the two variables can better be affected by an intervention are not valid. This question cannot be answered with the data at hand. As stated above, this topic should be omitted.
End of Abstract
‘Conclusions. We believe the GSES represents a useful tool to examine the efficacy of proposed and existing diabetes self management, and management self-efficacy interventions.’
I cannot understand that the results imply that GSES represents a useful tool to examine the efficacy of proposed and existing diabetes self management.

Additional comments

nothing

·

Basic reporting

Language
Line 28: alter ‘self management’ into ‘self-management’
Line 32: the official abbreviation of the General Self-Efficacy Scale is GSE and not GSES
Line 33: alter ‘Diabetes-management’ into ‘Diabetes Management
Line 40: alter ‘diabetes-management’ into ‘diabetes management
Line 43: alter ‘self management’ into ‘self-management’
Line 78: alter ‘Self-efficacy’ into ‘self-efficacy’
Line 82: alter ‘Bijl, J. van der et al., 1999’ into ‘van der Bijl et al., 1999’
Line 89: alter ‘Diabetes’ into ‘diabetes’
Lines 93 and 94: alter ‘Diabetes management self-efficacy (Diet-, Exercise-, Monitoring-
and Medical treatment-self efficacy)’ into ‘diabetes management self-efficacy (diet-, exercise-, monitoring- and medical treatment self-efficacy)’
Lines 94 and 95: alter ‘General and Diabetes management self-efficacy’ into ‘general- and diabetes management self-efficacy’
Line 114: alter ‘Indonsians’ into ‘Indonesian’
Line 153 and 154: alter ‘Diet self-efficacy, Exercise self-efficacy, Monitoring self-efficacy
and Regimen self-efficacy’ into ‘diet self-efficacy, exercise self-efficacy, monitoring self-efficacy
and regimen self-efficacy’
Line 169 to 172: not relevant for the present study ‘Other psycho/clinimetric instruments included the Summary of Diabetes Self-Care Activities developed by Toobert and colleagues Toobert et al. (2000), and the Diabetes Knowledge Scale developed by Beeney and others. Beeney L. et al. (2003). We don’t consider these scales in the present study, but leave this for future work.’
So these sentences can be deleted.
Line 185: alter ‘Parallel’ into ‘parallel’
Line 196: alter ‘Bartlett?s’ into ‘Bartlett’s’
Line 213: alter ‘Monitoring self efficacy’ into ‘monitoring self-efficacy’
Title Figure 1, before line 239: alter ‘Corrgram of General- and Diabetes management-self efficacy’ into ‘Corrgram of general- and diabetes management self-efficacy’
Line 239: alter ‘General and Diabetes-management’ into ‘general and diabetes-management’
Title Table 3: alter ‘General self efficacy and Diabetes management’ into ‘general self-efficacy and diabetes management’
Line 250: alter ‘General and Diabetes management’ into ‘general and diabetes management’
Line 259 and 261: alter ‘Diabetes’ into ‘diabetes’
Line 269: alter ‘Diabetes-management’ into ‘diabetes management’
Line 279: alter ‘(e.g. Monitoring Self-efficacy)’ into ‘(e.g. monitoring self-efficacy)’
Line 280: alter ‘Regimen DMSEs’ into ‘(regimen DMSES)’
Line 287: alter ‘Exercise-, Nutrition- and smoking abstinence- self-efficacy’ into ‘exercise-, nutrition- and smoking abstinence self-efficacy’
Line 292: alter ‘General and Diabetes management’ into ‘general and diabetes management’
Line 337: alter ‘self efficacy’ into ‘self-efficacy’
Line 338: alter ‘metabolic-syndrome’ into ‘metabolic syndrome’

Abstract
In the background nothing is written about the motive for this study: the lack of a validated instrument to measure GSE.
In the methods section is not described what reliability and validity measures are studied and the sample size included in the study is not mentioned.
Specific results about the reliability and validity measures are not reported, e.g. Cronbach’s alpha.
The main conclusion, that the Thai GSE is a reliable and valid measure, is missing.

Experimental design

Methods
Is the convergent validity meant with the association of patient characteristics with both GSES and DMSES in line 203? If yes, mention explicitly the convergent validity. If no, what kind of validity measure is referred to.

Results
Table 1. Education: the different forms of education in percentage don’t add up to an exact 100%.

Validity of the findings

Okay

Additional comments

Compliments to the authors. They accurately addressed the comments of the reviewers. Some minor and last remarks are made to finish the article for successful publication.

---

## Round 0.3 · Minor Revisions

Dear Dr. Rakkapao,

Your manuscript entitled "Psychometric properties of the General Self-Efficacy Scale among Thais with Type 2 diabetes, and its relationship with patients' disease management self-efficacy: A multicenter study " has again been carefully reviewed by the Editor and Reviewers. Basically the revision is now acceptable for publication, but before final acceptance is given, I would appreciate it if you would address the remaining minor issues raised by Reviewers 1 and 3.

If you are willing to do this, it would not be necessary for me to return the manuscript to the reviewers, but it could then be accepted for publication. Please track and indicate the changes made to facilitate the revision of the new version.

Sincerely yours,
Stefano Menini

Reviewer 1 ·

Basic reporting

Lines 171-172

‘Patient characteristics were summarized using means and standard deviations, for continuous variables, and counts and percentages for categorical measures.’

My comment in the last review
‘This sentence is poorly formulated. Very confusing.’


The authors’ reaction
‘Sorry, we have to disagree. This sentence is very clear and is almost a copy and paste from the hundreds of other ‘statistical analysis’ sections that I have written in the past. It is grammatically correct and in no way convoluted. It simply states that the appropriate statistics were used for patient characteristics measured using different measurement scales (means and standard deviations for continuous variables, and counts and percentages, for categorical variables). I am not assuming that it is correct simply because it has been acceptable in the past, but simply because it is a very common way of phrasing which statistics are used in the ‘descriptive statistics’ subsection of a results section (very often Table 1). Not only do I teach my students (as an academic biostatistician) to state how they use their summary statistics in this way, but this is the way almost all researchers in the clinical sciences are taught to do this.’

My reaction to the authors’ reaction
‘I stick to my claim that there is a linguistic flaw in the sentence. Simply leave out the comma between ‘deviations’ and ‘for’, then the sentence becomes clear.’





Sub-chapter statistical analyses (Lines 170-200)

In my last review, I made to comments regarding this sub-chapter. Unfortunately, in my last review the reference of the first comment to this sub-chapter has been lost when rearranging my original review according to the PeerJ-Guidelines. Therefore, I want to repeat them now with explicit reference to this sub-chapter:

1) ‘The authors should divide the whole sub-chapter into more than one paragraph.
2) Before describing a statistical procedure, the authors should mention the purpose for which this procedure is applied. Especially, this includes giving a clear overview of the criteria applied for judging validity. Still more especially, this includes some information they have provided in their comments to my previous review.
3) A part of the text within this sub-chapter already refers to results. The authors should formulate their methods part so that there is no need to refer to results and present the results in the results part.’


The authors comment to my first comment was that they could not assign this comment to text in their manuscript. They were right. Now they can.

The authors reply to the second comment was:
‘We do acknowledge that our statistical analysis section is methodologically dense, but
beyond providing details one might find in a psychometrics or statistics textbook we had already done exactly what the reviewer is suggesting (providing the reasoning for each analytical step). Furthermore, we have not really ‘left the beaten track’ of what represents standard psychometric validation practice.
In the statistical analysis section, every method we use is in explained in terms of its use, and where appropriate, criteria (e.g. cut-offs) are provided for judging validity. Specifically:

1. Internal consistency reliability - > Cronbach’s alpha > 0.8
2. Determining the number of factors -> Exploratory factor analysis with parallel analysis
3. Test the structural validity of GSE in the Thai T2D population -> A one factor measurement model based on an unweight least squares confirmatory factor analysis -> successfully fits (=>structural validity) if TLI, CFI and AGFI >0.9 and RSMEA<0.08
4. Further evidence of construct validity based on the Kaiser-Meyer-Olkin and Bartlett’s test of Sphericity
5. Divergent validity (GSE from DMSES) based on Disattenuated correlation coefficient < 0.85
6. Convergence between GSE and DMSES (overall scale and subscales) based on Pearson’s correlation coefficient
7. In what manner GSE and DMSE associate with (distribute across) important patient characteristics -> Ordinal logistic regression.’

My comment to the authors’ comments
I try to rephrase my former comment. I would have liked to read some sentences as to how the construct of general self-efficacy, i.e. the construct to be measured, implies the validity criteria used by the authors. In other words, I would have liked to read some sentences why and how the statistics applied by the authors reflect the extent to which GSE measures general self-efficacy and thereby are criteria for investigating validity. There are no such sentences. The whole sub-chapter does not convey the impression that the authors have thought much about problems of this kind. Accordingly, there is much statistics, but only little relevant knowledge produced by using this statistics.


The authors reply to the third reply was

‘Actually the only time we mention ‘results’ in our methods section is when they come from other studies, and these results are used to justify our methodological choices. However, we see in one particular case there is ambiguity (the last sentence of the DMSES subsection):

“Sangruangake et al. (2017) not only showed that DMSES was valid in Thai adults with T2D, but the overall scale and several of its subscales were highly predictive of concurrent HbA1c control. For instance, the overall DMSES was shown to have a sensitivity and specificity for predicting HbA1c control of 81% and 84%, respectively.”

We see some of the results of the Sangruangake et al. (2017) were split over two sentences, but the reference was only included in the first. We have now reiterated the citation in the second sentence:

“Sangruangake et al. (2017) not only showed that DMSES was valid in Thai adults with T2D, but the overall scale and several of its subscales were highly predictive of concurrent HbA1c control. For instance, the overall DMSES was shown to have a sensitivity and specificity for predicting HbA1c control of 81\% and 84\%, respectively (Sangruangake et al., 2017).”’

My comment to the authors’ reply
I am sorry. My comment was too unspecific. The sentence in the methods section that refers to own results is ‘ As the parallel analysis demonstrates a single factor had an eigenvector significantly greater than 1 (see Supplementary Figure 1),…’.

Experimental design

Background section in the abstract; lines 29-32
For example, it’s unclear whether self-efficacy is modifiable through intervention, or whether it is a relatively stable personality trait. A way of investigating this is to examine the nature of the relationship among the domain-specific Diabetes Management Self-Efficacy Scale(DMSES), and the arguably more stable general self-efficacy, as measured by the General Self-Efficacy scale(GSE).

The comment that I made in the last review regarding this topic was

‘The interrelation between two variables gives no information as to which of these two variables can be better affected by an intervention!’
The authors should omit all considerations into this direction!

The authors’ reply was:
‘We take the reviewers point here. However, to suggest that all research is purely empirical and that researchers should never speculate about the downstream implications of their findings (or the findings of others) unless those findings have already been proven elsewhere is likely to severely constrain what can be done. Speculation where (most) readers can agree on the plausibility of explanations is a very important part of research, especially research concerned with psycho-social elements of health.

It is clear the reviewer does not like words like ‘logical’, however, it seems much more likely that an intervention involving disease-specific education and disese-specific self-care practice (for example) is likely to impact disease-specific self-efficacy

Finally (and perhaps most importantly), we don’t make the claim that the inter-relationship between GSE and DMSE tell us which is better effected by an intervention. It is important to note that several RCTs have been run (by others) to gauge the efficacy of various diabetes management self-efficacy programs, and we are in no way (and at no time) recommending that interventions be designed to improve. Instead, we suggest that it is possible that the efficacy of a diabetes management self-efficacy intervention (which are currently run by others) on individual patients may be explored in terms of that patient’s general self-efficacy. These two objectives (or statements) are worlds apart.’

My comment to the authors’ reply
‘Regarding the first paragraph: I do think that speculation plays an important role in science. Among other things, it can be used to derive research questions and, thereby, to instigate empirical research. However, it should not be used as a substantiation of an empirical method. This is what the authors implicitly do in lines 29-32.

Regarding the second paragraph: I do not see that this reply relates to my comment.

Regarding the third paragraph: In lines 29-32, the authors do make the claim that the inter-relationship between GSE and DMSE tell us which is better effected by an intervention.

Conclusion: I stick to my comment that the paper would improve if the authors omitted all parts of the manuscript in which they suggest that the research presented in their paper gives any information as to which of the two variables can better be affected by interventions.

Validity of the findings

.

·

Basic reporting

Language
Line 31: insert a space between Scale and (DMSES)

Line 32: insert a space between scale and (GSE)

Line 51: alter ‘self management’ into ‘self-management’

Line 89: alter ‘Bijl, J. van der et al.,’ into ‘van der Bijl et al.,’

Line 91: alter the dot after ‘circumstances’ into a space

Line 110: delete ‘Luszczynska et al. (2005)’

Line 113: alter ‘treatment-self efficacy’ into ‘treatment self-efficacy'

Lines 133 and 134: alter ‘In our study, 749 patients who had been diagnosed with T2D within the last five years were...’ into ‘In our study 749 patients, who had been diagnosed with T2D within the last five years, were...

Line 139: alter ‘Scale’ into ‘scale’

Line 162: alter in both cases ‘Scale’ into ‘scale’

Line 177 and 178: ‘along with many other laboratory measurements’. This is not relevant. Mention only those characteristics which are reported in Table 1.

Line 215: alter ‘characteristic’ into ‘characteristics’. This is also the case at the top of Table 1. And check if the patient characteristics in Table 1 are all consequently presented in capitals. For example see ‘Waist Circumference’ and ‘Marital status’

The marginal note of Table 1: insert a space between ‘n’ and ‘(%)’, and also between ‘mean’ and ‘(SD)’

Line 237: alter ‘diabetes-management’ into ‘diabetes management’

Lines 239 and 240: both lines start with ‘Figure 1’. Delete the second Figure 1 and continue line 239 with ‘and’

Line 248: alter ‘General and Diabetes-management' into ‘general and diabetes management’

Line 250: insert a space between ‘Table 3’ and ‘shows’

Table 3: check if the patient characteristics in Table 3 are all consequently presented in capitals, and at the head of the Table alter ‘General self efficacy’ and ‘Diabetes management self efficacy’ into ‘General self-efficacy' and ‘Diabetes management self-efficacy'

Figure 1: alter ‘diabetes management-self efficacy’ into ‘diabetes management self-efficacy', and alter 'scale’ into ‘scales’

Line 281: insert ‘a’ between ‘be’ and ‘strong’

Line 347: alter ‘Hypertension and Dyslipidemia’ into ‘hypertension and dyslipidemia’

Experimental design

No comments

Validity of the findings

No comments

Additional comments

Some minor language remarks have been made as the last step for publication.

---

## Round 0.4 · accepted · Accept

Thank you for submitting a revised version of your manuscript entitled "Psychometric properties of the General Self-Efficacy Scale among Thais with Type 2 diabetes, and its relationship with patients' disease management self-efficacy: A multicenter study ".

I am pleased to inform you that your manuscript is accepted for publication in PeerJ in its current form and will now be forwarded to the product editor for publication.

I thank the reviewers for their effort in improving the manuscript and the authors for their cooperation throughout the review process.

Sincerely yours,
Stefano Menini